# The Use of Fluorides in Public Health: 65 Years of History and Challenges from Brazil

**DOI:** 10.3390/ijerph19159741

**Published:** 2022-08-08

**Authors:** Paulo Frazão

**Affiliations:** Department of Politics, Management and Health, Public Health School, University of São Paulo, São Paulo 01246-904, SP, Brazil; pafrazao@usp.br

**Keywords:** public policy, health policy, sanitation policy, fluorides, health surveillance, dental public health

## Abstract

Untreated dental caries is the most common disease globally and fluoride use at the population level is crucial for its control. To investigate the economic and political conditions under which the trajectory of population-based fluoride use has occurred is key for a more comprehensive view on its current and future challenges. The objective was to give a brief history and summarize the information on the use of fluorides at the population level in Brazil from 1952 to 2017 and to point to current and future challenges. A critical overview was undertaken based on country-level analyses comprising political and economic conditions. The analytical approach adopted a set of premises applicable to the study of capitalist democracies. Fluoride methods of systemic and topic use began to be employed in Brazilian public health programs in the 1950s and in a combined way from 1974. Differences in political and economic contexts were highlighted for four periods: the first interventions from 1952 to 1974, when the fluoridated water law was approved; the expansion after 1974 until 1988, when a new constitution was enacted; the following time until 2010; and the final period. From the 1980s to 2008, water fluoridation coverage increased progressively, consolidating as a major strategy of systemic use in spite of inequalities among territories. Activities aimed to promote access to topical fluoride use increased and maintained stability until 2014, when they dropped sharply. Regulation of fluoride dentifrice’s quality remained insufficient. It was hypothesized that the strengthening of conservative liberalism and the increase in fiscal austerity observed in recent years might produce serious constraints on public investment and limit access to fluorides. To reduce inequities and promote benefits for all, including the most vulnerable groups, policies based on egalitarian and social justice theoretical perspectives are needed more than ever.

## 1. Introduction

Untreated dental caries in the permanent dentition is the most common disease globally. A significant proportion of the population suffers pain, infection, tooth loss, loss of school days and decreased productivity and other impacts on oral health and quality of life [1]. The observed decline in its prevalence in high-income countries, and in some middle-income countries such as Brazil, could erroneously lead to the conclusion that there is no longer any need to maintain public health strategies aimed at preventing and controlling dental caries. However, the disease is still considered a priority on the public health agenda [2], and the methods recognized as the most effective in its prevention involve the use of fluorides [3]. Regions where the population has easy access to fluorides, such as some countries in Latin America, continue to show a downward trend in the occurrence of the disease [4].

The most effective public health strategies in preventing caries are applied at the population level, where exposure to fluoride is regular and sufficient to maintain constantly low concentrations in the oral cavity by cost-effective strategies [5]. Its use in public health has reached millions of people around the world, covered through specific programs financed by the state, and also by regulatory arrangements involving the market for fluoride products and dental services. Public health strategies may vary from country to country depending on specific environmental, health, political and economic conditions. To summarize the trajectory of population-based fluoride use according to these conditions and characteristics related to policies of sanitation and health sectors is key for a more comprehensive view on its current and future challenges.

Brazil has a long history of fluoride use in public health that has gone through different sanitation and health policies under diverse political and economic contexts to reach the current status of a multiparty capitalist democracy similar to many countries. Although this upper-middle-income country has experienced a phase of growth and reduction of inequalities at the 21st century’s beginning, its position in the ranking of social indicators remains far below in relation to its position in the ranking of economic indicators: in 2020, Brazil ranked 12th place in relation to gross national income and 81st in relation to life expectancy at birth [6]. 

The purpose of this study was to give a brief history and summarize the information on the use of fluorides at the population level in Brazil from 1952 to 2017 and to point current and future challenges. 

## 2. Materials and Methods

A critical overview of major analyses on the trajectory of public policies directed to increase the access to fluoride at the population level was undertaken. The scientific information was selected and its main features were reported. Links and comparisons were made where relevant, with analyses of both sanitation and health sectors. Preference was given for country-level analyses comprising political and economic conditions. 

The official data sources for estimates of the coverage for treated water have different origins but usually came from demographic census and from public and private companies responsible for service delivery. The data from these sources were checked. Coverage of fluoridated and treated water took the number of households connected to water networks as a reference. Surveys and analyses signed by researchers experienced in the subject were also consulted. Whenever possible, preference was given to values provided by careful and accurate analyses recognized by the scientific community through peer-reviewed publications. As data required for estimating the coverage of activities with topical use of fluoride were not available, the monthly mean of activities in relation to the target population was used to calculate the outputs needed to achieve results. The target population was defined as the total population from 5 to 9 years of age, estimated for the country in each year. All indicators were based on official data sources, and major investigations had used these data.

The analytical approach adopted a set of premises applicable to the study of capitalist democracies in which the direction of the state and the allocation of public funds is a permanent matter of disputes between different perspectives. At one side, a more liberal trait in which the market plays a prominent role and the state must restrict itself to functions not provided by the former, and the other, in which the state has an essential role in shaping the functioning of markets for certain goods and services and reducing the adverse effects owing to capitalism [7]. From an ethical–political point of view, these perspectives are divided into several categories that can be condensed among those who believe that individual interest has precedence over general interest and those who recognize the relevance of the general interest over that of the individual. These convictions guide different visions of democracy and rights. While the first usually supports the civil and political rights within a formal democracy, the second defends democracy also as an economic category in which social rights are assured for each citizen independently of income or ability to pay [8]. The results are presented below.

## 3. Results

The estimates were organized and presented through six moments Figure 1, taking into account seven references [9,10,11,12,13,14,15]. The values showed a progressive expansion of treated water coverage between 1970 and 2017. In the last three decades of the 20th century, coverage increased by around 35 percentage points (p.p.), while in the subsequent period the increase was around 10 p.p. In relation to water fluoridation, the expansion was progressive until 2008, when it stopped at around 76%.

As the data required for estimating the coverage of activities with the topical use of fluoride were not available, the ratio between the monthly mean of activities and the target population was estimated. Differently from an earlier picture, between 1994 and 2017, the ratio values took the form of an inverted parabola. After the creation of specific federal incentives provided by the Brazilian Unified National Health System for the financing of dental programs aimed to ensure access to topical fluoride use, the ratio began to increase until 2004, reaching about 50% of the 5-9-y.o. population. This pattern, with small variations, was maintained for 10 years until 2014, when it began to drop sharply (Figure 2).

The discussion of the results is structured in four sections, referring to the first interventions from 1952 to 1974 when the water fluoridation law was approved; the expansion after 1974 until 1988, when a new constitution was enacted; the following time until 2010; and the final period. This division was built taking into consideration differences in political and economic contexts, changes in sanitation and health sectors, and the availability of data on the interventions and the selected analyses.

## 4. Discussion

### 4.1. First Interventions

The use of fluorides in public health started in 1952 in Brazil by means of activities in dental care school programs conducted by the Public Health Special Service (*in Portuguese Serviço Especial de Saúde Pública—SESP*). This federal government agency was created in 1942 from an agreement between the governments of Brazil and the United States of America, having as functions the sanitation of regions which produced primary material, such as rubber from the Amazon region and the iron ores and mica from the Vale do Rio Doce, in the state of Minas Gerais. It was a major organization for the development and dissemination of knowledge and technologies for dental public health in the country. As a measure for dental caries prevention at the population level, 2 percent sodium fluoride solution was applied on schoolchildren’s dental surfaces by specially trained dental hygienists, and the four-chair system was recommended to increase the coverage of the program [16].

In that period, studies on naturally occurring fluoride in surface and groundwater sources were being undertaken [17]. As evidence of the relationship between fluoride, both with fluorosis and with caries, was being accumulated, awareness grew about the importance of knowing the natural levels of fluoride, which is still reflected currently in many publications related to Brazil [18] and other regions of the Earth [19].

The first device to adjust the fluoride concentration in the public water supply was installed in the municipality of Baixo Guandu, in the state of Espírito Santo, in the southeastern region of the country in 1953. The first results of dental caries decline due to adjustment around values of fluoride concentration in water between 0.7 and 0.9 mg F/L were published in 1957 [20]. Since then, other initiatives have succeeded. Between the 1970s and 1980s, studies on the intervention’s effects in the cities of Campinas, Piracicaba, Bauru, Uberlândia and Barretos, all in the southeastern region of the country, were published. Reduction in dental caries prevalence was observed in the permanent teeth of schoolchildren, measured by the DMFT index after the implementation of the fluoride concentration adjustment device in the water supply system [21,22,23,24,25]. The reduction values were similar to those observed in the first studies carried out in the USA and Canada [26]. The expansion of the measure in the USA and the approval by the 22nd World Health Organization (WHO) Assembly, held in Boston, in the United States, in July 1969, of the recommendation to member countries to examine the possibility of its introduction in public supply systems with waters below optimal levels, collaborating for the implementation of the first systems in Brazil [9].

According to Figure 1, data compiled until 1972 indicated that fluoridation systems already reached more than three million inhabitants, which corresponded to 10% of the population covered by treated water [9,11]. Of those systems, more than half were located in three states: Minas Gerais, Santa Catarina and Rio Grande do Sul, as a result of the encounter between favorable local conditions and the strategies adopted by SESP. In 1952, the federal agency organized the first Autonomous Water and Sewage Service in a municipality of state of Minas Gerais. In those first twenty years, the federal agency played a very important role in the trajectory of sanitation, by proposing the creation of local entities with full legal, administrative and economic autonomy, enabling the training of Brazilian technicians abroad and the first public health courses for engineers and disseminating modern techniques of filtration, disinfection and water fluoridation [27].

With the military coup in 1964, the first bill proposal for the fluoridation of drinking water presented one year earlier in the Deputy Chamber was archived. A 25-year cycle of development, in which the country’s essentially rural economy specialized in the production of a few tropical agricultural commodities for export, had been transformed into a recognizable industrial economy, with nearly half the population concentrated in urban areas, began to change. In that period, Brazilian industry appeared to have attained enough size and a sufficient degree of internal diversification to set the nation on the course of self-sustained economic growth [28].

The military government considered the lack of water supply and sewage as the main housing problem. Due to economic reforms undertaken, sanitation measures gained traction. A sanitation financing fund managed by a public company for supporting housing construction was created in 1967. The strategies were to encourage the creation of sanitation companies owned by the state-level governments under a concession regime and to provide water and sewage treatment services within a model that foresaw long-term support through a tariff regime [29]. Within that context, a national sanitation plan was implemented, the results of which are described in the next section.

The first interventions with fluorides were characterized by the adoption of a single method of prevention. Combinations of topical and systemic fluoride administration methods as a population strategy would be developed only in the following period.

### 4.2. The Expansion after 1974

The first proposal of a national bill on fluoridation in the water public supply was inscribed in 1963, but only in 1974 Law 6050 which determines the fluoridation of water in supply systems where there is a treatment plant, was approved by the National Congress. The previous year saw the apex of the dispute between the hard and moderate segments that commanded the military dictatorship, and the year 1974 represented the beginning of a period of slow and gradual distension towards the restoration of democratic processes in the country in response to the economic crisis that was felt as a result of the growth model that was being adopted and the shock caused by oil prices at the international level [30]. Oral health advocacy coalitions comprising professional and academic leaders and the support of federal agencies obtained adherence among deputies. With the enactment of the regulations that followed Law 6050 the measure gained public policy status [31]. Further, the country’s redemocratization process constituted the framework under which its progressive expansion would take place (Figure 1).

A significant increase in the population’s access to treated and fluoridated water happened in the 1970s and 1980s. Data from the period 1970–1972 to 1990–1995 showed that the coverage of treated water increased from 34.0 to 72.0%, while the coverage of treated and fluoridated water increased from 3.4 to 42.2% (Figure 1). Detailed analysis showed that the increase occurred in all areas of the country. Although part of the inequalities between geographic regions, the demographic size of the municipalities and the monthly income of the households had been reduced [11], territories in the North and Northeast regions and population groups that lived in precarious housing remained without access to treated water at the end of the 1990s [32].

An important factor in the increase in coverage was the creation of the Sanitation Financial System in 1967 and the National Sanitation Plan in 1971. Despite the evolution of the indicators, a strong business concept prevailed in which investments were oriented towards works that allowed the generation of greater profitability, providing a faster return on invested capital [33]. The Plan proved to be more strongly focused on the construction and expansion of systems than on aspects of their operation. The low operational efficiency came out as a serious problem. The loans taken by the companies owned by the state-level governments, high interest rates, difficult capital market conditions and growing interest expenses led them to a tough situation. So, from the 1990s onwards, the Plan was abandoned as it no longer responded to the challenges of the expansion and qualification of the sector [29].

Another aspect refers to efforts maintained by the SESP, which came to be called the National Health Foundation and whose management model was based on administrative decentralization through a municipal autarchy, led to the creation of more than 2300 autarchies up to the mid-1990s [34].

Regarding fluoridation, the observed expansion can be associated with different political and economic aspects in the period. From an economic point of view, the difficulties in obtaining the fluoridating compounds used in the first years, in which high prices, due to the need to import such products, greatly hampered the dissemination of the measure [9], were gradually being overcome with the development of technologies using fluosilicic acid, a liquid effluent whose internal availability increased from the 1980s onwards as a result of the growth in the manufacture of phosphoric acid necessary for the fertilizer industry. Brazilian consumption of fertilizers increased from 958 thousand tons in 1970 to around 5 million tons in 1995 [35].

Another aspect concerns the creation of lines of financing. Soon after the approval of Law 6050/1974, water fluoridation became an official health policy and an agreement was signed between the SESP and the National Institute of Food and Nutrition, which started to incorporate fluoridation among the projects aimed at combating specific nutritional deficiencies. Another initiative was federal support through the financial and housing construction support agencies, creating conditions for resources from the social investment fund that could be used to pay for the implementation of new fluoridation systems [36].

Furthermore, at the political level, while the military regime was weakening, the struggles for the country’s redemocratization multiplied. Brazil had about 7000 expatriated people and approximately 800 political prisoners. Between 1974 and 1985, a law was passed granting amnesty to all those who committed political or electoral crimes and to those who suffered restrictions in their political rights. In 1980, the 7th National Health Conference, and in 1986, the 1st National Oral Health Conference were held, in which the importance of expanding water fluoridation was reaffirmed. Direct elections for governors and for the most populous Brazilian municipalities were resumed in 1982, and some local governments began to set up innovative projects for primary healthcare. This process of redemocratization created conditions for the renewal of ideas and political practices, of unions, parties, scientific associations, professional entities, churches, and above all, for the emergence of grassroots social movements in which actors of different origins acted in the defense of citizenship and elementary rights, such as drinking water, sanitation and health [33,37]. During the struggle to restore democracy, a social movement in favor of health reform took place, assembling service providers, workers and users. Many principles claimed by the movement would be added to the Brazilian Constitution of 1988 [38,39].

Together with the definition of a federal financing line with resources from a social investment fund, this process favored, at that time and in the following years, the expansion of basic sanitation, including water fluoridation, where this specific measure was claimed in a more articulated way by the different social actors [31]. However, this expansion faced barriers. Beyond those linked to specific points within the sanitation sector, such as insufficient financial support, the precarious physical structure of certain water treatment plants and the unavailability of fluoridating compounds, in the late 1980s, inspired by the salt fluoridation results disseminated by the Pan-American Health Organization (PAHO) and supported by salt producers, professional groups began to propose the implementation of fluoridation in the country [40]. Although such a proposal had obtained support from the Brazilian Health Ministry in 1990, the technical/scientific debates that followed this initiative led several major health departments at state and municipal levels to oppose the measure implementation, which did not prosper in the country [41].

In addition to water fluoridation spreading, this period was marked by major innovations in school-based caries preventive programs. At the end of the 1970s, the topical application of 2 percent sodium fluoride solution through the four-chair system was replaced by a mouth rinse program using 0.2% sodium fluoride solution at least at every two weeks. Several studies had demonstrated its effectiveness [42,43], with the advantage of requiring a minimal effort for beneficiary compliance and having low personnel requirements.

In the 1980s, the method, quickly spread by several dental school programs, was being developed in the country and managed by different organizations such as SESP (federal agency) and education departments at the state and local levels [44]. However, the high dropout and school failure rates [45], combined with high social mobility as a result of rising housing prices in metropolitan areas, made it difficult to assess their results, and then other topical methods requiring only two applications per year were proposed [46]. Both total and soluble fluoride concentrations in Brazilian toothpastes have been evaluated since the early 1980s. Until 1988, only 25% of toothpastes sold in the market were fluoridated [47].

### 4.3. After the 1988 Constitution

Between 1990–95 and 2008–10, the coverage of treated water increased almost 10 percentage points and the coverage of fluoride-adjusted water increased by about 34 percentage points (Figure 1). The population recovered the right to vote for president in 1989. In this 20-year period, the country was managed by center-right coalitions between 1990 and 2002 and by center-left coalitions between 2003 and 2010.

Given the crisis that befell the sanitation framework of the earlier period, a new orientation based on efficiency was sought for the sector during the 1990s. Some programs were tried with the aid of multilateral bodies during the center-right coalitions. A significant part of Brazil’s budget was allocated in debt service payments and public sector investment was restricted. From 2003 and 2010, the federal government tried to go ahead with the possibility of public/private partnerships (PPP), but ownership disputes between federated entities, regulatory instability and unavailability of long term credit lines for financing new infrastructure projects represented important hurdles [29]. As a consequence, the coverage increased lightly but failed to reach full coverage in the urban water service. A new regulation mark was approved in the end of this period that is described in the next section.

Regarding fluoridation, the important observed expansion took place in a context of much legislative interest [40] and also of implementation of municipality-funded programs for water fluoridation surveillance [48]. Water fluoridation coverage tends to be higher in larger municipalities [49] with higher human development levels [50,51]. Although the sanitation model in Brazil did not constitute an example of equity [52], water fluoridation coverage from 2000 to 2008 showed major reductions in absolute and relative between-municipality inequality according to population size and human development levels [13].

Seven bill proposals were presented on fluoride systemic administration methods in which four enforced the salt fluoridation and three revoked the fluoridation law approved in 1974 [40]. All of them were discussed, rejected and filed in the most important deliberative space of the country. Analyses under distinctive approaches of these parliamentarian debates showed disputes between different political and economic perspectives of social development [40,53,54].

Since the 1980s, researchers had showed the need for fluoridation monitoring as a means to assure effectiveness concerning tooth decay prevention [55,56]. As there is no fluoride-free water, the WHO recommended that public health surveillance agencies pay proper attention to the fluoride parameter. The surveillance carried out by bodies that are not responsible for the treatment and distribution of water was the main strategy to ensure its quality. This can be done by auditing the operational control data generated by the companies or by directly collecting samples at strategic points in the distribution network [57].

In Brazil, Vasconcellos [58] noticed that the occurrence of dental caries among 7-to-12-year-old schoolchildren who were born and always resident in the city was more frequent than expected for a community in which public water supply was fluoridated for 16 years, demonstrating that discontinuity of fluoride concentration at optimal levels had occurred in the early 1970’s. During the military dictatorship, political and civil rights were suppressed; books were censored and the funds for research dwindled. As scientific activity became scarce, only several years later did another study show an unexpected increase in caries severity during the military government in areas where the measure had been implemented, also suggesting discontinuity [59].

Once effect from water fluoridation on dental caries prevention could only be measured some years after its implementation, Brazilian specialists have recommended the surveillance to be carried out by organizations not directly responsible for the water treatment (the principle of external control) by means of the direct evaluation of water samples collected at the distribution system so as to assure the quality of the process, information validity and reliability for reaching oral health goals. Studies based on this premise showed that municipality-funded surveillance programs led to positive effects on the fluoridation quality of the public water supply [12,60,61].

After the year 2000, a normative device began to require municipal health authorities to implement an appropriate sampling plan to guide surveillance actions [62]. An information system was developed by the Brazilian Ministry of Health for supporting the National Program of Surveillance on Environmental Health Related to the Quality of Water for Human Consumption. Analysis from a nationwide study showed that 37.3% of Brazilian municipalities fed the system of the fluoride parameter for at least four months in 2008, confirming the initial stage of implementation [63].

In the late 1980s, two important changes contributed to spreading the access to topical fluoride methods. The first was the approval of health as a right for all and a duty of the state in the 1988 Constitution and the creation of the Unified Health System (*in Portuguese Sistema Único de Saúde—SUS*), so called for breaking with the existing division between the preventive activities maintained by the Ministry of Health and the individual dental and medical care activities funded by the Ministry of Social Security. Under the principles of health services integration and decentralization, the legislation and other regulating devices were enacted and the majority of units for specialized and primary healthcare maintained by these federal agencies were progressively transferred to states (26 units more the federal district) and large municipalities [64].

The second point refers to fluoride dentifrices reaching the market on a large-scale basis in 1989, and the federal agency responsible for health surveillance published requirements that these products should fulfill. With this achievement, Brazil ranked third regarding per capita toothpaste consumption, behind the United States and Japan [47].

In 1992, the federal government began to encourage the implementation of preventive programs, transferring financial resources for local governments. For this, a set of disease prevention and oral health promotion activities, of low complexity, dispensing dental equipment should be developed, aiming to reach population subgroups such as children from a daycare center, students from a preschool, clients enrolled in a basic health unit or other groups in certain social spaces that could be accompanied during the program. This set comprised annual dental examination in order to measure changes in the oral health profile of the group served, quarterly educational activities and at least one topical fluoride method (weekly mouthwashes with 0.2% sodium fluoride solution and/or daily supervised brushing with 1000 ppm fluoride dentifrice) [64].

The process of health services integration and decentralization advanced and 5343 (97.0%) municipalities were formally qualified to manage the services in 1999. To finance primary health care, the resources came to be transferred to the municipalities in a per-person amount [65]. The number of municipalities without dental care resources decreased. The majority of dental equipment located in schools was transferred to the primary health care network, thereby creating the conditions for greater integration between dental care activities and other health programs [64]. The schools were no longer a place to carry out individual clinical and surgical-restorative activities and started to provide programs for disease prevention and health promotion.

Time-series analysis related to the volume of these activities showed a consistent increase until the year 2000 and a stationary trend between 2001 and 2007 [66]. Despite differences among municipalities [67] and temporal variations in the same municipality [68], this program was gradually expanding and occupying a prominent place in local oral health policies across the country. In the 1990s, the highest figures were found in four states: Mato Grosso, Mato Grosso do Sul, Minas Gerais and Santa Catarina [69]. A study in this last state, from 2000 to 2003, showed that the magnitude of these activities was negatively correlated with the proportion of tooth extractions in relation to procedures of individual primary oral healthcare [70]. Although the specific federal incentives had been subsumed in general financial transfer strategies to meet local level demands for more flexibility since 1999, no study showed a reduction in the provision of these activities in the country as a whole (Figure 2).

In 2006, the Brazilian Health Ministry started to require the registration of the oral disease prevention and health promotion activities in a discriminatory way [69]. Six procedures were encoded, (a) collective action of supervised tooth brushing with fluoride dentifrice; (b) collective action of fluoride mouthwash; (c) collective action of topical application of fluoride gel (d) collective action of oral examination with an epidemiological purpose; (e) collective health education activity by a higher education professional in the community; (f) collective activity by a mid-level professional in the community. The last two were not exclusive to the oral health set. Researchers found an increase in supervised tooth brushing activities with fluoride dentifrice between 2006 and 2010 [71].

Another point to highlight was the creation of the Health School Program in 2007. As one of its goals was to expand disease prevention and health promotion activities under the principles of comprehensiveness, territoriality and intersectorality, it would be expected for the next period to have some impact on the delivery of topical fluoride use.

Low-cost toothpastes are usually formulated with calcium carbonate (CaCO_3_) as an abrasive and sodium monofluorophosphate (Na_2_FPO_3_; MFP) as an anticaries active component, and depending on the storage time, the MFP ion becomes chemically inactive [72]. A negative aspect was that the regulation requiring toothpastes had a minimum soluble fluoride concentration after manufacturing and after one year, was abandoned in 1994, and since then Brazilian health surveillance agency has not fulfilled its responsibility for ensuring that the population has access to good-quality fluoride toothpastes with anticaries potential. As generally these low-cost toothpastes are those most used in public health programs and by families of lower socioeconomic status (75% of the Brazilian population depends on publicly funded health services), it is possible that the expected effects due to the use of fluoride toothpastes are not being fully reached.

### 4.4. From 2010 to 2017

Between 2008–2010 and 2017, the coverage of treated water and the coverage of fluoride-adjusted water had little variation (Figure 1). Data related to 2008 showed that about 57.7% of municipalities had services provided by a company managed by state or federal government or an association between public entities, and the remaining ones were companies managed by the municipality or an inter-municipal consortium (25.0%) and by an association between public and private entities (13.5%) and by a unique private company (3.8%). Among municipalities, 39.8% did not have fluoridation services. The lack of fluoridation provision was higher when the service was provided by municipal administrations and private companies, whether or not associated with public entities and regardless of the characteristics of the municipalities [73].

In the end of the earlier period, the parliament had approved law 11,445 which defined the universal access to water and sewage services as a fundamental principle. Known as the basic sanitation regulatory framework, the law defined the municipality as the holder of the title to sign contracts with public or private companies. Strengthening the decision-making power of the municipal entity proved to be more advantageous for large municipalities that had larger budgets and more developed basic services than for small municipalities that had reduced budgets and difficulties in authorizing tariffs incompatible with the ability to pay of their populations. However, the water supply sector remained in need of federative coordination, financial conditions, harmonized and balanced rules and legal confidence for providers, where the perception of the involved risk continued to be high, making public and private companies’ participation unattractive [74]. This picture would be faced with the edition of a new law in 2020 that is outside the scope of this article.

Regarding water fluoridation, the given priority in the earlier period seemed to have finished. As there is a dependence relationship between the implementation of public policy and the existence of water treatment plants and water supply networks, it is recognized that the provision of this public policy could be greater in Brazil because almost all municipalities were supplying water by means of a general supply network in at least one of their districts or part of it [75]. Thus, it would be plausible to investigate the reason why fluoridation provisions did not reach more municipalities, given that the cost of the measure was reduced when a water treatment and supply structure existed and that the efficiency of the service was not affected [76,77]. Moreover, it has proven to be effective even with the widespread use of fluoridated toothpaste [78]. A study based on the theory of street-level bureaucracy investigated small towns in a high-coverage region of the country. Institutional characteristics such as the administrative fragility of local entities, low priority given locally to policy, poor physical structure of the water treatment plants, isolated working relations, low effectiveness of monitoring devices and local actors’ uncertainties about the policy favored the expansion of the discretionary power of street-level operators, configuring important barriers for water fluoridation [79].

The pace of expansion observed in previous periods seems to have reached a saturation point. To overcome this situation will require the resumption of a proactive agenda aimed at adjustments in ongoing initiatives and review of incentives for opening space in the agendas of local health and sanitation policies. Study in a Brazilian state located in the Central–West region of the country concluded that improving the use of existing policy coordination mechanisms for the effective implementation of water fluoridation was clearly needed and that partnership creation, consolidation and shared mission, especially among health and sanitation sectors, are challenges to be faced by policy implementation [80].

Some innovative traits of Brazilian history have been the surveillance policy based on external control data [12]; the promotion of technical consensus for determining the range concentration values of best risk-benefit combination [81]; and the formulation of quality indicators [61]. Several publications have used these criteria. Investigating the surveillance system of drinking water quality in a high-coverage Brazilian state, researchers showed that 56.6% municipalities had very good fluoridation quality, e.g., 80% of samples in each municipality were within the optimal level range (0.55 to 0.84 mgF/L). To ensure the high quality of the public health strategy, additional management measures should be implemented in municipalities with less than 100,000 inhabitants, with a higher chlorine concentration nonconformity rate, with a lower per capita income and where the type of sanitation utility is municipal or private [82]. On the other hand, a nationwide study showed that the implementation of surveillance systems remained uneven among federative units. Approximately 134,000 records were reviewed. Of the Brazilian municipalities, 1810 (32.5%) had valid information for the fluoride parameter, with substantial variation between the South (83.6%) and North (0.7%) macro-regions. Of these, 726 (40.1%) showed very good fluoridation quality with higher values (54.3%) in municipalities with 50,000 inhabitants or more, and lower values (34.2%) in those with less than 10,000 inhabitants [83].

In relation to the activities of disease prevention and health promotion, two studies confirmed reductions for the country as a whole [71,84]. Examining these yearly, the number of activities was stationary until 2014 when it began to decline sharply (Figure 2). In that year, President Dilma Rousseff was reelected by a narrow margin of votes, and conservative and ultra-liberal sectors began to see a gap to weaken the center-left coalition through an anti-corruption crusade driven by an alliance between segments of the judiciary and the great media. With the replacement of the elected president that was boosted by a center-left coalition with a vice-president supported by a center-right coalition, fiscal austerity measures were implemented. In health, there was a reduction in the number of health professionals, a reduction in access to medicines, and worsening of health indicators [85].

The measures of fiscal austerity supported by the center-right coalition were associated with the deterioration in the numbers of disease prevention and health promotion activities [86,87]. Regulation of fluoride dentifrice quality remained insufficient [72].

Since 2009, data from national surveys of school health have shown that one in three Brazilian students in the 9th grade had insufficient frequency of daily tooth brushing to reach the oral health goals [88]. In relation to the School Health Program (SHP) created in the earlier period, actions have failed to reach 20% of adolescents enrolled in each school. One study showed the health sector’s leadership but peripheral performance from the education sector. School health activities had a biomedical approach and were carried out through lectures. The SHP seemed to have strengthened the relationship between the two sectors. However, aspects of the intersectoral articulation in the political-management process and practices presented weaknesses and limitations [89].

## 5. Conclusions

A critical overview of 65 years of fluoride use in public health was presented. Fluoride methods of systemic and topical use began to be employed in Brazilian public health programs in the 1950s and in a combined way from 1974. Between the 1980s and 2008, the water fluoridation coverage increased, progressively reaching three-quarters of the population and consolidating as major strategy of systemic use in spite of inequalities among the North and Northeast regions compared with the remaining ones. In the second decade of the 21st century, about 25 countries maintained water fluoridation schemes in the world [5], and only 7.6% of people in middle-income countries were benefitting from fluoride in drinking water, while the figure counted 20.9% for high-income countries [90].

Activities aimed to promote access to topical fluoride use increased and maintained stability until 2014, when they dropped sharply. Data from 101 countries measured 10 years after the 2007 WHO Assembly statement on oral health indicated that 3 in 4 countries had oral health school programs, though these were less frequent in low-income countries. One-third of them offered fluoride mouth rinsing in schools for children aged 5–7 and 12 years [90]. A global survey by the WHO in 2012 indicated that health education, supervised tooth brushing and fluoride use were the most common school-based strategies [91].

Demanding a good level of quality of interventions in Brazil has been a requirement of diverse sectors, including the academy. Major innovative traits in Brazilian history have been the surveillance policy based on external control data; the promotion of technical consensus for determining the range concentration values of best risk–benefit combination; and the formulation of quality indicators. In spite of these efforts, the implementation of surveillance systems of drinking water quality remained uneven among federative units, and the regulation of fluoride dentifrice quality has been kept insufficient.

It is fascinating to note how these strategies were engendered under distinctive political and economic conditions: from the 1950s when the country passed by a democratic period with industrialization and economic growth; the 1960s and 1970s, related to the apogee and decline of the autocratic period with economic growth and inequality increases; and the redemocratization period within which welfare state policies and neoliberal state policies have been alternated [92,93].

Both strategies expend a very low fraction of total cost of water treatment services and of total cost of publicly funded dental services and their expansion and delivery at optimal level have been subject to political and economic conditions. While economic conditions refer to financial support for sanitation and health policies, political conditions refer to the ability of stakeholders in these public health strategies to make governments of different levels and supporting political coalitions prioritize them on the agendas of the health sector. In relation to water fluoridation, an additional challenge refers to the sanitation sector.

Brazil is a multiparty capitalist democracy with a three-level federal system that assures power and relative autonomy for the central government (first level), 26 states and one Federal District (second level), and 5570 cities (third level). Federal republics with autonomous and inter-dependent governmental levels face more difficulties than countries with unitary governments for accomplishing goals from policies to promote equity [7].

The configuration of sanitation service provision reflects past policies and present neoliberal trends, bringing specific management characteristics to each location in the country, with important implications for the provision of fluoridation. Since 2007, local levels have held the prerogative to manage the sanitation in their jurisdiction, including changes and renewal of contracts with service providers. If water is not fluoridated, judicial and political institutions must be prompted to require it from sanitation companies. Therefore, it is important to update oral health advocacy coalitions articulated with general health demands and progressive sectors of society at all government levels.

At the federal level, seven bill proposals were presented in the studied period: four enforced the salt fluoridation and three revoked the fluoridation law approved in 1974. All of them were rejected in the most important deliberative space of the country. Democratic achievements of Brazilian society as the water fluoridation Law can be lost whether the positive points cease to prevail over the negatives invoked in the debate on population-based health policy.

At state levels it is important to adopt procedures to spread and improve the surveillance system of drinking water quality implementation, and at local levels it is crucial to support activities aimed to implement the public health strategy where it was interrupted or has not been implemented yet. There is great concern in countries with a federative structure in relation to the mechanisms for coordinating public policies so that they are actually fulfilled and not just “on paper”. This fact is not only a problem for health policies but affects all public policies in Brazil [94].

In relation to the health sector, local levels have also held the prerogative to manage it in their jurisdiction and need financial support for this. However, measures of fiscal austerity supported by center-right coalitions have been associated with the deterioration in numbers of disease prevention activities at the end of the analyzed period. Such measures have been used as responses to economic crises and fiscal deficits in both developed and developing countries [95].

In conclusion, the trajectory of water fluoridation coverage was always upward, while the activities aimed to promote access to topical fluoride use were marked by a rise, a hold on a plateau and a sudden decline, meaning that the second seems to be more vulnerable to economic and political changes than the first. A possible hypothesis for this difference between the trajectories would be that the water fluoridation has been favored by the legacy of sanitation policies and the political deliberations based on right to health endorsed by the 1988 Constitution, which has supported the maintenance of the water fluoridation law. The legacy of sanitation policy led to the creation of state companies enabled to take the institutional lead in the provision of water services in Brazil, with power to allocate resources, making them solely responsible for setting priorities and selecting technologies and expansion strategies [96]. These structural features explain the robustness of sanitation policy to withstand the innovations and the economic and political changes.

On the other side, although topical fluoride use had been disseminated owing to the specific federal incentive policy based on right to health endorsed by the 1988 Constitution and to the health policy strategies aimed to health services decentralization and integration, these specific incentives were clustered in general financial transfer strategies to meet local level demands for more flexibility. As the provision of the activities promoting access to topical fluoride use began to drop up to 2014, with the economic and political crisis that had, as a result, the replacement of a center-left coalition by a center-right one, a hypothesis is that the fiscal austerity measures implemented by this coalition led to the reduction of financial transfers for local levels compromising the delivery of these activities.

If the Brazilian history between 1964 and 2019 can be seen as a parabola in which the militaries with power to dictate the rules had to leave the scene to the redemocratization movement, and the economic and political crisis after 2014 saw a gap to colonize power, once again challenging progressive forces, which were trying to defend the democratic frameworks established by the 1988 Constitution pacts [93], the trajectory of topical fluoride use between 1988 and 2017 equivalent to an inverted parabola could to be associated with a growing curve of democratization that falls back under the rise of fiscal austerity supported by a center-right coalition.

A general hypothesis that emerges from this overview is that the strengthening of conservative liberalism and the increase in fiscal austerity observed in recent years might produce serious constraints on public investment and limit access to fluorides. To reduce inequities and promote benefits for all, including the most vulnerable groups, policies based on egalitarian and social justice theoretical perspectives are more needed than ever.

## Figures and Tables

**Figure 1 ijerph-19-09741-f001:**
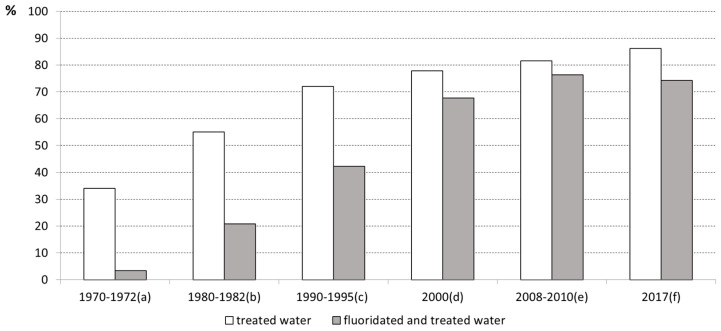
Population covered by treated and fluoridated water through public supply systems in different periods from 1970 to 2017. **Sources**: (**a**) [9,11]; (**b**) [10,11]; (**c**) [11,12]; (**d**) [11,13]; (**e**) [13,14]; (**f**) [15].

**Figure 2 ijerph-19-09741-f002:**
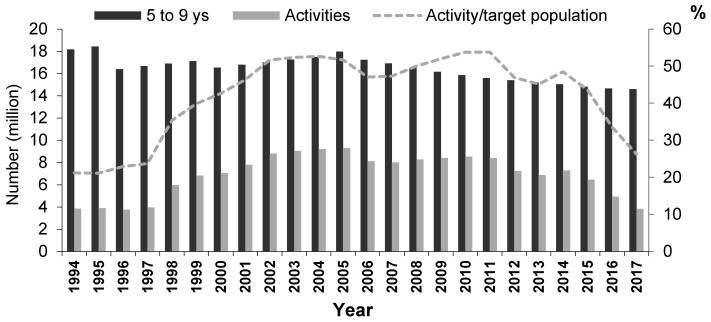
Monthly mean of activities with topical fluoride use provided by publicly funded health services in relation to the estimated total population from 5 to 9 years old. Brazil, 1994–2017. **Source**: Organized by the author based on data extracted from the Ambulatory Information System provided by Informatics Department of Brazilian Unified National Health System (http://www.datasus.gov.br, accessed on 21 February 2022). **Note:** The number of registered activities since 2006 is the sum of three actions: (a) collective action of supervised tooth brushing with fluoride dentifrice (code 01.01.02.003-1); (b) collective action of fluoride mouthwash (code 01.01.02.002-3); (c) collective action of fluoride gel topical application (code 01.01.02.001-5) (http://sigtap.datasus.gov.br/tabela-unificada/app/sec/inicio.jsp, accessed on 21 February 2022).

## Data Availability

Data sources were described in the text.

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
