# Peer review of "The Use of Fluorides in Public Health: 65 Years of History and Challenges from Brazil"

_ijerph, 2022, doi:10.3390/ijerph19159741_

Round 1

Reviewer 1 Report

This article deals with important issue and important analysis for public health policy analysts and practitioners.

1. The article is too long for journal readers. It seems that it can be re-phrased without loss of data.

2. I would be beneficial if a published, validated methodology of public policy analysis would be described in Material and Methods.

3. Conclusions section includes Discussion issues, and Summary besides conclusions derived from the results. 

Author Response

I thank the reviewer for your thoughtful and in-depth comments concerning the manuscript.

  1. The article is too long for journal readers. It seems that it can be re-phrased without loss of data.

The first version of the text before the submission was twice bigger. After successive revisions for summarizing the content, the submitted version was reached. Taking into account the reviewer’s suggestion, additional effort was made to identify and remove minor aspects, and rewrite sentences. Among several changes, the content connected to the ref. 91 was excluded. Therefore, the words’ number was reduced in the resubmitted version.

  1. I would be beneficial if a published, validated methodology of public policy analysis would be described in Material and Methods.

As the reviewer well observed, the manuscript deals with important analysis for public health policy analysts and practitioners. As it was mentioned in the Methods, the selected scientific literature was summarized based on an overview in which peer-reviewed investigations on trajectory of public policies directed to increase the access to fluoride at Brazilian population level were included. Preference was given for country-level analyses comprising political and economic conditions. Some selected studies have a clear quantitative approach while others have a qualitative approach in which the data were provided by interviews and by documentary research and the observations’ interpretation was based on different political science theories such as public policy cycle; street-level bureaucracy; and historic institutionalism. As a result of these aspects an overview was elected as the more adequate approach to summarize the 65-year trajectory of investigated public policies.

  1. Conclusions section includes Discussion issues, and Summary besides conclusions derived

The Results’ section holds only the quantitative description of selected indicators while the Discussion’s section summarizes and condenses the main analyses. The Conclusions’ section is more long than usual because it holds the main points related to the public policies’ trajectory, the challenges and related hypothesis for each one of them. The first paragraph of Discussion’s section reports on the access to water fluoridation while the second refers to access to topical fluoride in the studied case. Third paragraph points to the contributions of Brazilian history. From forth to tenth paragraph I summarized the challenges as advertised in the title, and in the last four paragraphs I built some hypotheses based on the literature presented in the overview. I believe that all these points meet the overview’s objectives and can be extremely useful for readers interested in the relationships between political/economic conditions and public health policies.

Reviewer 2 Report

Thank you for the opportunity to review this manuscript entitled “The use of fluorides in public health: 65 years of history and challenges from an upper-middle-income country.” The article aimed to give a brief history and summarize the information on use of fluorides at population level in Brazil from 1952 to 2017 and to point to current and future challenges. I think that this manuscript was well presented by the author, and its content may be of interest to the Brazilian readers. However, I'm not sure if readers from other countries get interested in your article. Because I felt like I was reading a public health textbook in Brazil.

I have some comments below:

1) It is recommended to add "in Brazil" to the title.

2) I think you can move the part of the first paragraph in Results section to Materials and Methods section.

3) How about adding the latest data, for example, 2021?

4) I think that the shorter conclusions are better for readers.

Author Response

I thank the reviewer for your thoughtful and in-depth comments concerning the manuscript.

The manuscript refers to a 65-year overview on health public policies aimed to increase the access to fluoride at population level in an upper-middle-income country. All researchers who investigate the relationships between political/economic conditions and public health policies surely will be interested in the manuscript’s content. Majority of public policy studies refer to case studies that hold a more comprehensive view on the current characteristics and future challenges of a determined public policy within a given context.

1) It is recommended to add "in Brazil" to the title.

As suggested, I added the country’s name in the title.

2) I think you can move the part of the first paragraph in Results section to Materials and Methods section.

As suggested, the content was displaced.

3) How about adding the latest data, for example, 2021?

The period after 2017 is not in the study’s scope because the last national survey on basic sanitation gathered data for 2017. From 2017 to 2021 no survey brought accurate data on coverage of treated and fluoridated drinking water. At 2020 the demographic census was not carried out in the country. After 2019, the Severe Acute Respiratory Syndrome Coronavirus 2 (SARS-CoV-2) emerged in Wuhan, China and rapidly spread to other countries, starting a pandemic period that required mitigation measures of virus transmission such as stay-at-home and school closure (on-site learning). Analysis of pandemic’s consequences for public health policies aimed to promote access to topical fluoride and to fluoridated drinking water at level population is matter to other further studies.

4) I think that the shorter conclusions are better for readers.

The Conclusions’ section is more long than usual because it holds the main points related to the public policies’ trajectory, the challenges and related hypothesis for each one of them. The first paragraph reports on the access to water fluoridation while the second refers to access to fluoride topical use in the studied case. Third paragraph points the contributions of Brazilian history. From forth to tenth paragraph I summarized the challenges as advertised in the title and in the last four paragraphs I built some hypotheses based on the literature presented in the overview. I believe that all these points meet the overview’s objectives and can be extremely useful for readers interested in the relationships between political/economic conditions and public health policies.